# Immunogenicity, Efficacy, and Safety of a Novel Synthetic Microparticle Pre-Erythrocytic Malaria Vaccine in Multiple Host Species

**DOI:** 10.3390/vaccines11121789

**Published:** 2023-11-30

**Authors:** Thomas J. Powell, Jie Tang, Robert Mitchell, Mary E. DeRome, Andrea Jacobs, Naveen Palath, Edwin Cardenas, Michelle Yorke, James G. Boyd, Stephen A. Kaba, Elizabeth Nardin

**Affiliations:** 1Artificial Cell Technologies, Inc., 5 Science Park, Suite 13, New Haven, CT 06511, USA; jtang@artificialcelltech.com (J.T.); mary.derome@yahoo.com (M.E.D.); ajacobs@artificialcelltech.com (A.J.); naveen.palath@pfizer.com (N.P.); ecardenas@artificialcelltech.com (E.C.); myorke@artificialcelltech.com (M.Y.); jboyd@artificialcelltech.com (J.G.B.); 2Department of Microbiology, School of Medicine, New York University, New York, NY 10010, USA; ramitchell2@gmail.com (R.M.); nardin.elizabeth@gmail.com (E.N.); 3Multiple Myeloma Research Foundation, 383 Main Avenue, 5th Floor, Norwalk, CT 06851, USA; 4Pfizer, Inc., Andover, MA 01810, USA; 5Malaria Vaccine Branch, Walter Reed Army Institute of Research, Silver Spring, MD 20910, USA; stephen.kaba@greenlightbio.com; 6GreenLight Biosciences, Inc., Lexington, MA 02421, USA

**Keywords:** malaria vaccine, microparticle, non-human primate, peptide, sporozoite

## Abstract

We previously reported a protective antibody response in mice immunized with synthetic microparticle vaccines made using layer-by-layer fabrication (LbL-MP) and containing the conserved T1BT* epitopes from the *P. falciparum* circumsporozoite protein. To further optimize the vaccine candidate, a benchtop tangential flow filtration method (LbL-by-TFF) was developed and utilized to produce vaccine candidates that differed in the status of base layer crosslinking, inclusion of a TLR2 ligand in the antigenic peptide, and substitution of serine or alanine for an unpaired cysteine residue in the T* epitope. Studies in mice revealed consistent superiority of the Pam3Cys-modified candidates and a modest benefit of base layer crosslinking, as evidenced by higher and more persistent antibody titers (up to 18 months post-immunization), a qualitative improvement of T-cell responses toward a Th1 phenotype, and greater protection from live parasite challenges compared to the unmodified prototype candidate. Immunogenicity was also tested in a non-human primate model, the rhesus macaque. Base layer-crosslinked LbL-MP loaded with T1BT* peptide with or without covalently linked Pam3Cys elicited T1B-specific antibody responses and T1BT*-specific T-cell responses dominated by IFNγ secretion with lower levels of IL-5 secretion. The Pam3Cys-modified construct was more potent, generating antibody responses that neutralized wild-type *P. falciparum* in an in vitro hepatocyte invasion assay. IgG purified from individual macaques immunized with Pam3Cys.T1BT* LbL-MP protected naïve mice from challenges with transgenic *P. berghei* sporozoites that expressed the full-length PfCS protein, with 50–88% of passively immunized mice parasite-free for ≥15 days. Substitution of serine for an unpaired cysteine in the T* region of the T1BT* subunit did not adversely impact immune potency in the mouse while simplifying the manufacture of the antigenic peptide. In a Good Laboratory Practices compliant rabbit toxicology study, the base layer-crosslinked, Pam3Cys-modified, serine-substituted candidate was shown to be safe and immunogenic, eliciting parasite-neutralizing antibody responses and establishing the dose/route/regimen for a clinical evaluation of this novel synthetic microparticle pre-erythrocytic malaria vaccine candidate.

## 1. Introduction

The latest estimates provided by the World Health Organization show an increase in malaria cases worldwide, from 230 million in 2015 to 247 million in 2021, while the number of malaria deaths worldwide rose from 577,000 to 619,000 over the same time period [1]. Africa accounts for 90% of all cases and deaths worldwide, and *P. falciparum* is the causative agent of nearly all malaria cases in sub-Saharan Africa. Current methods of malaria control are limited to vector control and post-infection therapeutics, but these have limited utility due to sporadic use, environmental concerns, and the emergence of resistance in both the vector and parasite. The development of widely useful vaccine approaches has been hampered by the complex life cycle of *P. falciparum*, which is characterized by immunologically distinct profiles in the vector, pre-erythrocytic, hepatic, and blood stages. Pre-erythrocytic vaccines are designed to neutralize infectious sporozoites immediately after injection into the host by an infected mosquito taking a blood meal or by interrupting the life cycle at the hepatic stage by killing infected liver cells before the release of merozoites that infect erythrocytes and lead to clinical malaria [2,3].

Validation of the pre-erythrocytic vaccine approach was provided by studies showing that immunization with attenuated sporozoites elicited protective immunity in multiple species including humans [4]. Antisera and lymphocytes from patients immunized with attenuated sporozoites were used to identify the circumsporozoite (CS) protein as a major protective antigen for pre-erythrocytic vaccines. These results have been further validated with the most advanced pre-erythrocytic vaccines in development, including irradiated sporozoites (Sanaria, Rockville, MD, USA) and the recombinant protein virus-like particle RTS,S (Mosquirix^®^, Glaxo Smith Kline, Brentford, UK), both of which express CS epitopes. In spite of promising results, these approaches are limited by logistical barriers associated with the scale-up and distribution of irradiated sporozoites [5,6,7], the requirement of complex adjuvants for RTS,S [8,9], and suboptimal sterile immunity that is short-lived in both cases [10,11,12,13] . Thus, there remains a need for a technology platform capable of delivering CS-derived epitopes to elicit potent, sterilizing, long-lasting immunity in the absence of complex adjuvants and with manageable logistical challenges.

We previously reported the application of layer-by-layer deposition to create synthetic microparticles displaying the dominant epitopes of CS, namely, the repeat B-cell epitope (NANP)_3_ that is the target of sporozoite-neutralizing antibodies, the CD4 helper epitope T1 that is conserved in all *P. falciparum* strains, and the T* epitope that is recognized by many HLA Class II molecules. Those studies showed that LbL-MP loaded with T1BT* designed peptides could elicit specific antibody and T-cell responses and protect the murine host from challenges with PfPb, a transgenic *P. berghei* expressing the CS subunits of *P. falciparum*. Protection was shown to be antibody-mediated, as alternative candidates that included only T-cell epitopes failed to elicit protection in the immunized mice [14]. However, T-cells are necessary to achieve the optimal potency of CS-based vaccines, as the induction of high levels of sporozoite-neutralizing antibodies is T-cell-dependent, requiring CD4+ T-cell help for B-cell differentiation [15] and for the development of CD8+ T-cells capable of killing infected hepatocytes and blunting the life cycle before progression to the blood stage [16]. T-cell cytokines such as IFNγ can also directly mediate protective immunity by inhibiting the development of intracellular exoerythrocytic forms [17,18]. Thus, the inclusion of conserved and universally recognized T-cell epitopes in CS vaccine design is warranted.

We now report the improvement of LbL-MP displaying the T1BT* subunit of CS. Several chemical modifications were introduced to the vaccine design, including crosslinking of the base layer polypeptide film to increase stability and potency, the inclusion of a TLR2 ligand to engage the innate immune system and improve the adaptive immune response both quantitatively and qualitatively, and replacement of an unpaired cysteine with serine to avoid inter-chain disulfide bond formation and simplify manufacturing. Compared to the basic design, vaccine candidates incorporating one or more of these modifications elicited higher titer parasite-neutralizing antibody responses that persisted for 18 months, broader antibody isotype distribution, a Th1-biased T-cell response, and more potent protection from *Plasmodium* challenges in the mouse model. The most promising candidates incorporated base layer crosslinking with or without the inclusion of the TLR2 ligand and were tested in a non-human primate (NHP) model. The results showed that the TLR2 ligand-modified candidate elicited the most potent antibody responses that conferred protection against parasite challenges upon passive transfer to naïve mice. The final modification of cysteine → serine substitution yielded a vaccine candidate that was easier to manufacture, retained immunogenicity and efficacy in mice, demonstrated a large therapeutic index, and elicited parasite-neutralizing antibodies in a Good Laboratory Practices compliant toxicology study in rabbits. The current results set the stage for a clinical evaluation of the safety and immunogenicity of this novel LbL-MP vaccine in healthy malaria-naïve adults.

## 2. Materials and Methods

### 2.1. LbL Particle Fabrication

CaCO_3_ microparticle cores were prepared by rapid mixing of equal volumes of 0.33 M Na_2_CO_3_ (containing 1.2 g/L poly-l-glutamic acid [PGA]) and 0.33 M CaCl_2_ in a flow reaction vessel. The resulting spherical porous CaCO_3_ microparticles containing a surface layer of PGA were collected in fractions throughout the flow process, and the particle sizes of individual fractions were measured by dynamic light scattering (DLS) using a Malvern Mastersizer 3000 particle analyzer (Malvern Panalytical, Boston MA). Fractions containing microparticles with a mean particle diameter (by volume) <5.0 μm and 90% of the particles <10 μm were pooled to yield a particle suspension containing approximately 1.6% solids in 0.33 M NaCl and excess PGA. LbL-MP were fabricated as previously reported [19] by alternately layering poly-l-lysine (PLL, positive charge) and PGA (negative charge) on the CaCO_3_ cores to build up a 7-layer base film and capping with an outermost layer of the designed peptide (DP). Where indicated (bXL), the PGA:PLL base film was chemically crosslinked by treatment with 200 mM EDC and 50 mM sulfo-NHS (Sigma-Aldrich, St. Louis MO, USA) in 0.2 M phosphate buffer, pH 6.5, for 30 min at room temperature prior to layering the DP. DPs were synthesized and analyzed by standard techniques [19]. The wild-type T1BT* DP sequence derived from *P. falciparum* CS protein is DPNANPNVDPNANPNV(NANP)_3_*EYLNKIQNSLSTEWSPCSVTSGNG*K_20_, where the T1 epitope is underlined, (NANP)_3_ is the B epitope, the *T** epitope is italicized, and K_20_ is a poly-lysine tail that drives the assembly of the soluble DP into the LbL film. Table 1 describes the DP used to make the LbL microparticles. In some DPs, the unpaired cysteine residue in the T* epitope was replaced with alanine or serine to eliminate the possibility of inter-chain disulfide bonds. TLR2 ligand Pam3Cys was added where indicated by manual coupling of Pam3Cys-OH (EMD Millipore) to resin-bound T1BT*K_20_ peptide in 4:1 N-methylpyrrolidinone/dichloromethane using 2-(1H-benzotriazol-1-yl)-1,1,3,3-tetramethyluronium hexafluorophosphate (HBTU) activation. The final architecture of all constructs was CaCO_3_:PGA:PLL:PGA:PLL:PGA:PLL:PGA:DP. PGA, PLL, and DP contents were measured by amino acid analysis (AAA), and endotoxin content was determined by the Limulus Amebocyte Lysate assay (#50-647U, Lonza, Walkersville, MD, USA) [19]. Final constructs for mouse and NHP studies were formulated to a DP concentration of 100 µg/mL in 10 mM HEPES, pH 7.0, 5% mannitol, 0.2% carboxymethylcellulose (CMC). The rabbit toxicology study samples were formulated to a DP concentration of 100 µg/mL in 10 mM HEPES, pH 7.0, 10% mannitol, 7.5% PEG-4000. Formulated suspensions were aliquoted, frozen, lyophilized, stored at 2–8 °C, then reconstituted to a suspension by the addition of water just prior to injection.

### 2.2. Animals and Immunizations

Female C57BL/6J mice, 5–8 weeks old at study initiation, were obtained from Jackson Laboratory and used for all mouse studies. Mice were immunized on the indicated days by administration of LbL-MP adjusted to deliver the indicated dose of DP via the footpad (f.p.) route. On the indicated days, some mice were sacrificed, and spleens were harvested and teased into single-cell suspensions. Following lysis of erythrocytes, the lymphocytes were restimulated with the indicated peptide in murine IFNγ and IL-5 ELISPOT plates as described below. On the indicated days, mice were bled by a retroorbital puncture; blood was collected in heparinized Natelson tubes and centrifuged to remove cells and isolate serum. Antibody responses were measured by ELISA and sporozoite neutralization assay (SNA) as described below. All mouse studies were conducted in accordance with the guidelines and approved protocols of the NorthEast Life Sciences Animal Care and Use Committee (IACUC).

Female rhesus macaques were pre-screened for seronegativity to *P. falciparum*, divided into groups of 4, and immunized by intradermal (i.d.) injection of 50 μg of LbL-MP vaccine candidate on the indicated days. Blood was collected 14 days prior to the first immunization and 14 or 28 days following each immunization, as indicated. Sera and peripheral blood mononuclear cells (PBMC) were prepared and stored frozen until ready for analysis. PBMC were restimulated with T1BT* or T* peptides in monkey IFNγ or IL-5 ELISPOT plates as described below. Serum antibody responses were measured in ELISA plates coated with T1B peptide and in the SNA, both as described below. All macaque studies were conducted in accordance with the guidelines and approved protocols of the Walter Reed Army Institute of Research IACUC.

A Good Laboratory Practices (GLP)-compliant toxicology study was conducted at Toxikon (Bedford, MA, USA) in accordance with their IACUC guidelines and approved protocols. Male and female New Zealand white rabbits (6 of each gender per treatment group) were immunized by i.m. injection of 50, 100, or 200 µg of LbL-MP on days 0, 28, 56, and 84. Rabbits were bled 7 days prior to the first immunization and 7 days following each immunization. Sera and peripheral PBMC were prepared and stored frozen until ready for analysis. PBMC were restimulated with T1BT* or T* peptides in rabbit IFNγ and IL-5 ELISPOT plates as described below. Serum antibody responses were measured in ELISA plates coated with T1B peptide and in the SNA, both as described below.

### 2.3. Antibody Assays

Antibody responses were measured by ELISA essentially as previously described [14,19]. Briefly, plates were coated with T1B or T1BT* peptide or full-length recombinant PfCS protein (Protein Potential, Rockville, MD, USA) as indicated (100 μL/well of 2 μg/mL), blocked, and washed before the addition of test antisera. After incubation and washing away unbound antibody, detection antibody was added (100 μL/well of peroxidase-labeled goat anti-mouse, anti-monkey, or anti-rabbit IgG (KPL, Gaithersburg, MD, USA in BSA Blocker). In some cases, isotype-specific detection antibodies were used. After incubation and washing, 1 Step Ultra TMB (Thermo Fisher Scientific, Waltham, MA, USA) was added (100 μL/well) and incubated for 1–3 min to allow color development. The reaction was stopped by adding 25 μL/well of 2M H_2_SO_4_, and the absorbance was read at a wavelength of 450 nm in a microplate spectrophotometer. All incubations were for one hour at room temperature.

Functional antibody responses were measured by their ability to inhibit sporozoite invasion of hepatocytes in the sporozoite neutralization assay (SNA) using *P. falciparum* sporozoites (Sanaria, Rockville, MD, USA) [20], following the basic protocol of the transgenic sporozoite neutralization assay (TSNA) as reported [21,22]. Briefly, test antisera were diluted as indicated and incubated with 20,000 sporozoites per well in RPMI on ice for 30 min and then added to a HepG2 lawn in a 48-well plate. Control wells contained HepG2 cells plus sporozoites (no serum, maximum signal) or HepG2 cells alone (no sporozoites, minimum signal). Cultures were incubated at 37 °C in 5% CO_2_ for 3 days with daily medium changes. After 3 days, RNA was isolated, and parasite 18S gene expression was measured by qPCR. Gene expression was normalized against human β-actin expression and then converted to % neutralization using the following formula:% neutralization = 100 × [1 − (serum + sporozoites + cells)/(sporozoites + cells only)]

### 2.4. T-Cell ELISPOT

T-cell responses were measured essentially as described [14,19]. Briefly, ELISPOT plates were coated with capture antibody specific for the cytokine and species of interest (mouse, monkey, or rabbit IFNγ or IL-5; all obtained from commercial sources including MABTECH AB [Cincinnati, OH, USA] and Cell Sciences [Newbury Port, MA, USA]), incubated, washed, and blocked. Mouse splenic lymphocytes or monkey or rabbit PBMC were added at 2 × 10^5^ cells/well along with T1B or T1BT* peptide (0.1 mL/well of 5 μg/mL). Plates were incubated overnight at 37 °C and washed, and species-specific biotinylated anti-cytokine antibody was added (100 μL/well at 2 μg/mL in PBS + 1% BSA). Plates were incubated for one hour at room temperature, washed three times, and 100 μL/well of streptavidin-alkaline phosphatase (R&D Systems, Minneapolis, MN, USA) was added. Plates were incubated for one hour, then washed three times before the addition of 100 μL/well of BCIP/NBT substrate (R&D Systems). Plates were incubated for 15 min in the dark to allow the formation of spots, and the reaction was stopped by rinsing the wells with water. Plates were air-dried overnight, and the spots were counted on an AID Viruspot Reader (AID, Strassberg, Germany).

### 2.5. Efficacy Measurement

Protection from parasite infection in the mouse model was measured as the reduction in liver-stage infection following challenges with PfPb [14], a recombinant *P. berghei* (mouse pathogen) carrying a transgene containing the entire repeat region of *P. falciparum* CS protein (human pathogen); antibodies to *P. falciparum* T1B protect mice from challenges with PfPb sporozoites [14,23]. Briefly, immunized and naïve control mice were challenged by exposure to bites of PfPb-infected mosquitoes (5–15 bites per mouse). Forty hours post-challenge, the mice were sacrificed, and livers were harvested and homogenized in TriReagent (Sigma) using a Polytron PowerGen 500 (Fisher Scientific). RNA was isolated using QIAgen Mini Prep Kit and converted to cDNA using iScript Reverse Transcription Supermix (BioRad, Hercules, CA, USA), both according to the manufacturers’ directions. Parasite burden was monitored by quantifying *P. berghei* 18S rRNA levels in qPCR [14].

Efficacy in the NHP model was measured by passive immunization of mice (PIM). IgG was purified from individual NHP sera and from pooled pre-immune sera. For each NHP IgG sample tested, eight naïve C57BL/6J mice received 0.5–1.0 mg Ig i.p. on days −2, −1, and 0 relative to the challenge by i.v. injection of 5000 infectious sporozoites of *P. berghei* transgenically expressing full-length PfCS (Tg-*Pb*/*Pf*CSP) [24,25]. On days 7–15 post-challenge, thin blood smears were taken and analyzed by Giemsa staining (minimum of 5 fields per smear) to identify the first day of parasitemia for individual mice. Any mouse that remained parasite-free on day 15 was considered to have sterile immunity conferred by the passive macaque immune Ig.

### 2.6. Statistical Analyses

Data were analyzed in GraphPad Prism 9.0 by a Kruskall–Wallis test with Dunn’s correction or one-way ANOVA with Tukey’s test. Where noted for small sample size, data were analyzed by Student’s *t*-test. *p* < 0.05 was considered statistically significant.

## 3. Results

### 3.1. Candidate Optimization and Potency in Mice

We previously reported that several LbL-MP vaccine candidates containing epitopes of the PfCS protein elicited parasite-neutralizing antibodies and protected mice from in vivo challenges with live *Plasmodium* [14]. Those results demonstrated improved vaccine activity when the LbL-MP contained the TLR2 ligand, Pam3Cys, covalently linked to the antigenic DP. There was also a trend toward improved potency when the homopolymer base layers of the LbL-MP were crosslinked prior to the addition of the DP. Based on those results, candidate optimization studies were designed to directly test the impact of base layer crosslinking and the Pam3Cys modification of the DP on vaccine activity. Thus, we designed a panel of constructs that incorporated both Pam3Cys and crosslinking modifications in all four permutations, as summarized in Table 1. ACT-1198 was the simplest design without either modification, while ACT-1199 contained the Pam3Cys modification of DP, ACT-1200 had crosslinked base layers without the Pam3Cys modification, and ACT-1201 contained both crosslinked base layers and the Pam3Cys modification. In all other respects, including the base layer architecture, particle diameter, sequence of the DP, and DP loading density, the four candidates were consistent. Groups of C57BL/6J mice were immunized with each of the four candidates on days 0, 21, and 42, by f.p. administration of particle suspensions adjusted to deliver 10 µg of DP (normalized for Pam3Cys content, where applicable). On day 49, immune responses were tested by ELISA and ELISPOT, both assays using the T1B peptide that includes both the antibody and CD4 epitopes. The T1B-specific IgG responses revealed an interesting pattern, with the simplest candidate (1198) being the least potent, either crosslinking (1199) or Pam3Cys (1200) increasing potency by approximately 30-fold, and the combination of both modifications (1201) resulting in a further 8-fold increase in potency (Figure 1A). In addition to the quantitative improvement in antibody responses, the Pam3Cys modification also resulted in a qualitative improvement as it elicited detectable IgG2c isotype that is associated with Th1 responses in C57BL/6J mice (Figure 1B); crosslinking did not impact the isotype distribution of the antibody response. Indeed, the T-cell profiles reflect this shift toward Th1 with inclusion of Pam3Cys in the vaccine candidate: Figure 1C shows a very weak T-cell response in the 1198-immunized group, a modest IFNγ/IL-5 response in the 1199-immunized group, a potent response in the 1200-immunized group with particularly high numbers of IL-5-secreting cells, and similarly high levels of IFNγ-secreting but much lower levels of IL-5-secreting cells in the 1201-immunized group. These results demonstrate that either modification (base layer crosslinking or the addition of Pam3Cys to DP) increased the potency of the vaccine, while the combination of both modifications resulted in the highest potency for antibody responses and the most favorable Th1-biased T-cell responses.

### 3.2. Persistence of Antibody Responses

The mice depicted in Figure 1 were maintained without further intervention or manipulation to allow for analysis of the persistence of vaccine-induced immunity. On day 552 (17 months after the day 42 boost immunization), the mice were bled and the antibody titers were measured by ELISA. While the 1198-immunized mice had low levels of circulating T1B-specific antibody, the other three groups had higher antibody titers on day 552 (Figure 2), with only minor decreases compared to day 49, demonstrating that either modification (crosslinking or Pam3Cys) significantly improved long-term immunity in vaccinated mice. The same sera were assayed for parasite-neutralizing activity in the SNA, and the results in Figure 3 correlate closely with the ELISA results: all groups had high neutralizing antibody activity on day 49 (7 days post-boost), but the 1198-immunized mice had barely detectable neutralizing antibody activity at day 552, while the other groups retained neutralizing antibody titers with only modest decreases from the earlier time point. While the combination of both modifications was sufficient to improve the persistence of immunity in the immunized mice, it is noteworthy that either modification alone resulted in a dramatic improvement in immune persistence. It is perhaps most striking and unexpected that base layer crosslinking alone was sufficient to account for long-lasting neutralizing antibody titers (compare 1198 to 1200). While we have not yet formally investigated any putative antigen depot effect, it is certainly possible that base layer crosslinking improves particle stability and, therefore, increases the time of antigen exposure to the host’s immune system, resulting in more robust immune responses. When combined with the innate immune system’s engagement by the Pam3Cys modification, this results in an optimized vaccine that elicits favorable humoral and cellular responses.

### 3.3. Immunogenicity and Efficacy in Non-Human Primates

Of the four constructs tested, ACT-1198 was clearly the least potent in the murine studies and was, therefore, eliminated from further consideration, while ACT-1201 (bXL, Pam3Cys) elicited the highest antibody titers and the most favorable Th1-biased T-cell responses and was, therefore, selected for further consideration. The two single-modification designs, ACT-1199 (Pam3Cys) and ACT-1200 (bXL), elicited equivalent ELISA antibody titers but slight differences in isotype distribution and in T-cell phenotype. Since there was no statistical difference in neutralizing antibody titers in the ACT-1199 (nXL, Pam3Cys) and ACT-1201 (bXL, Pam3Cys) groups in Figure 3, ACT-1200 (bXL) was selected for further consideration due to the relatively simpler processes required in the manufacture of base layer-crosslinked particles compared to the synthesis, purification, and layering of lipidated DP on particles. Thus, ACT-1200 (bXL) and ACT-1201 (bXL, Pam3Cys) were selected for testing in non-human primates (NHP, Rhesus macaques) to determine the necessity of including the Pam3Cys modification of the DP.

Eight malaria-naïve NHPs were bled 14 days before vaccination and divided into two groups of four that were immunized intradermally with 50 µg of ACT-1200 or ACT-1201 on days 0, 28, and 56. NHPs were bled 14 days following each immunization for the isolation of sera and peripheral blood mononuclear cells (PBMC). Pre- and post-immunization sera were analyzed in the T1B ELISA. Figure 4A,B shows that, while both constructs elicited T1B-specific antibody responses, ACT-1201 was more potent than ACT-1200. The same sera were diluted 1:50 and tested for neutralization of *P. falciparum* in the SNA. Control wells contained HepG2 cells plus sporozoites (no serum, maximum signal) and HepG2 cells alone (no sporozoites, minimum signal). The results in Figure 4C show that antiserum from a single 1200-immunized NHP (#09U009) had high Pf neutralizing activity, while antisera from all four 1201-immunized NHPs had high neutralizing activity. Although there is not a direct correlation between the ELISA data and the SNA data at the individual animal level, it is evident that both assays detect a pattern of greater potency in the animals vaccinated with ACT-1201.

Collectively, the ELISA and SNA results show that ACT-1201 elicited the most potent functional antibody response. However, the ultimate test of antibody-mediated efficacy is the PIM model, in which naïve mice receive IgG purified from the NHP and are then challenged with Tg-*Pb*/*Pf*CSP (transgenic *P. berghei* displaying the full-length PfCSP) and monitored for the development of blood-stage parasites. A preliminary PIM study showed that sera from the 1200-immunized NHP failed to protect the mice from the challenge, while sera from the 1201-immunized NHP did afford varying levels of protection (Appendix A), which is in general agreement with the SNA results. Thus, the ACT-1201 group was immunized a fourth time and bled 14 days later, and the sera were tested in ELISA and SNA while IgG was purified and administered to groups of naïve mice that were then challenged. The results in Figure 4D show a rebound in antibody levels two weeks following the fourth immunization, while Figure 4E shows that antisera from each NHP neutralized sporozoite infectivity at dilutions ranging from 1:50 to 1:6250. When the IgG was purified from each individual serum and passively administered to naïve mice, it protected the mice from parasitemia following the challenge with PfPb, with from 50% to 87.5% of mice per group remaining parasite-free up to 15 days post-challenge (Figure 4F).

Although the protective mechanism of a CS-based malaria vaccine is predicted to be predominantly sporozoite-neutralizing antibodies, T-cell responses may also contribute both by helping B-cell responses and by exerting cytotoxic effector activity against infected hepatocytes. Thus, cellular immune responses in the immunized NHPs were measured in PBMC collected from individual NHPs and restimulated in IFNγ or IL-5 ELISPOT plates. Figure 5 shows that both constructs elicited T* or T1BT*-specific T-cell responses that were predominantly IFNγ with lower numbers of IL-5 spots. These results contrast with the murine results in which each construct elicited both phenotypes, with ACT-1200 favoring IL-5 and ACT-1201 favoring IFNγ. ACT-1201 was selected as the prototype candidate based on consistently superior antibody responses, efficacy, and T-cell responses in both murine and NHP models.

### 3.4. Immunogenicity and Efficacy of LbL-MP with Substitution for Unpaired Cysteine Residue

The wild-type sequence of the T* epitope in ACT-1201 contains an unpaired cysteine thiol that is prone to inter-chain disulfide bond formation and thus presents challenges for the manufacture, purification, and handling of DP containing this sequence. This risk was mitigated by making conservative substitutions at this residue and preparing LbL-MP with the substituted DP, otherwise retaining the DP sequence incorporated in ACT-1201. The DP ACT-2246 contains a Cys → Ala mutation and ACT-2247 contains a Cys → Ser substitution, and both DPs also have the tyrosine in the C-terminal lysine tail deleted. A single batch of seven-layer LbL-MP (CaCO_3_:PGA:PLL:PGA:PLL:PGA:PLL:PGA without DP) was prepared and split into two batches, one of which was crosslinked while the other was not. Each sister batch was split so that each new DP (C → A, C → S) could be separately layered onto non-crosslinked and crosslinked base layers, yielding constructs ACT-1236, -1237, -1238, and -1239 as shown in Table 1. Groups of mice were immunized with individual construct, including ACT-1201 as a control, on days 0, 21, and 42, and bled on day 49 for antibody measurement by ELISA. The results show that all four new constructs were comparable to the ACT-1201 prototype, yielding ELISA titers that overlapped those elicited by ACT-1201 (Figure 6A) and isotype distribution profiles that were also indistinguishable (Appendix A). The only difference in activity among the new constructs was observed in the T-cell ELISPOT, which revealed that ACT-1236 and ACT-1237 (C → A) failed to elicit potent IFNγ or IL-5 responses, while ACT-1238 and ACT-1239 (C → S) elicited both phenotypes, albeit at slightly lower levels than the prototype ACT-1201 (Figure 6B). When the mice were challenged with PfPb and monitored for liver-stage parasite development, all four candidates performed comparably to ACT-1201 (Figure 6C). The dominant mechanism of protection for CS-based vaccines, particularly T1BT*-based vaccines in the C57Bl/6J mouse (which does not recognize T* as a CD8 target epitope) is antibody-mediated, so these results would suggest that either of the substitutions would be acceptable. However, since both helper and effector T-cell responses appear to play a role in protective immunity in the human host, and the C → A substituted LbL-MP failed to mount a T-cell response, the C → S substituted candidates ACT-1238 (nXL) and ACT-1239 (bXL) were selected for further development.

### 3.5. Persistence of Antibody Response in Mice Immunized with ACT-1239

The mouse model was used to examine whether crosslinking of the base layers may impact the persistence of immunity and efficacy following vaccination with LbL-MP candidates. Female C57BL/6 mice (33 animals/per group) were immunized with 10 μg of ACT-1238 (nXL, Pam3Cys) or ACT-1239 (bXL, Pam3Cys) by footpad injection on days 0, 21, and 42; control mice were mock-immunized with PBS via the same route on the same days. On day 49, each group of 43 mice was divided into one cohort of 3 mice and four cohorts of 10 mice for further treatments and assessments as shown in the timeline in Table 2.

On days 49 and 223 (7 and 181 days post-second boost, respectively) and day 349 (7 days post-third boost), mice were bled and sera were harvested for determination of antibody responses by ELISA. The results in Figure 7A show that both constructs elicited comparable levels of T1B-specific antibodies of both IgG1 and IgG2c isotypes, with ACT-1239 being slightly more potent, and that the response patterns remained consistent throughout the study with only a slight decrease in signal at day 223 that recovered following the third boost. Figure 7B shows that both constructs elicited similar IFNγ and IL-5 cellular responses to T1B peptide when assayed following the second boost. Ten mice per group were challenged with PfPb on each of days 56, 257, and 356 to monitor efficacy, and the results are summarized in Figure 7C. This is where the most striking difference between the two candidates becomes apparent. While both afforded comparable high levels of protection (average of 72–84% reduction in parasite burden within each group) immediately following the second boost (challenge day 56), this protective effect is almost completely erased by day 257 in the ACT-1238 group (7% average reduction in parasite burden with zero mice fully protected), while modest efficacy is still detected in the ACT-1239 group (54% average reduction in parasite burden with 3 of 10 mice fully protected). Following a third boost, the efficacy of both candidates rebounded to levels comparable to the original levels after the second boost.

The cumulative results show that ACT-1239 conferred greater protection in the short term and greater persistence of protection 6 months after the boost, while both constructs provided equivalent protection after a long-term boost. Based on these results, and in the context of the earlier results showing a modest advantage of base layer crosslinking (Figure 1, Figure 2, Figure 3 and Figure 4), ACT-1239 was selected as the final candidate due to superior antibody responses, T-cell responses, efficacy, and the elimination of the manufacturing risk by substitution of the free cysteine with serine.

### 3.6. Safety and Immunogenicity in GLP-Compliant Rabbit Toxicology Study

The general safety, toxicity, and immunogenicity of ACT-1239 were examined in a GLP-compliant rabbit study in preparation for a clinical evaluation. A preliminary non-GLP immunogenicity study showed that rabbits immunized i.m. with ACT-1239 mounted both antibody and T-cell responses for the DP epitopes, thus verifying the selection of rabbits as the species for toxicity testing. New Zealand White rabbits (*n* = 6/sex/treatment group) were immunized four times at 28-day intervals by i.m. injections of PBS or 50, 100, or 200 µg doses of ACT-1239, thus providing coverage for a clinical trial design of up to three administrations of up to 100 µg per patient.

For analysis of immunogenicity, rabbits were bled prior to the first immunization and on days 8 (post-prime), 36 (7 days post-first boost), 64 (7 days post-second boost), and 92 (7 days post-third boost), and sera were collected for the determination of PfCS-specific antibody titers by ELISA. The results show that ACT-1239 elicited low but detectable dose-dependent antibody responses following the first immunization (Figure 8A), with increasing titers from 50 to 100 µg after the second immunization and no further increase in the titer at the 200 µg dose (Figure 8B). Maximal titers were reached in all groups with no dose-dependent trend after the third and fourth immunizations (Figure 8C,D).

To determine the parasite-neutralizing antibody activity in the rabbits following vaccination, sera from 12 rabbits per dose level per time point (7 days post-1° and 7 days post-4°) were pooled, diluted (1:100, 1:1000, and 1:10,000), and pre-incubated with viable Pf sporozoites prior to the addition to HepG2 cell monolayers. The RNA was purified from the cells after 72 h and the Pf RNA was measured using qPCR. Figure 9 shows that all immune sera exhibited titratable neutralizing activity, and there was very little dose-response from the 50 to 200 µg doses, and the post-4° sera were more active than the post-1° sera from the same dose levels.

Only minor and sporadic clinical events were observed during the study. Given the low incidence of the adverse events, their transient nature, and the slight to moderate severity, none were considered toxicologically relevant. No animals were found moribund or dead during the study, and all animals survived until the end of the study with the exception of a single rabbit that was euthanized on day 58 due to a self-inflicted toe wound that was not treatment-related. There were no statistically significant differences in injection site irritation or food consumption and only transient differences in body weights that did not appear to be dose-dependent. Necropsy, hematology, clinical chemistry, and urinalysis revealed only sporadic and transient changes that were not statistically significant. The only exceptions were elevated fibrinogen and C-reactive protein (CRP) levels in some animals immediately following immunization that resolved to baseline levels within a few days, suggesting a mild inflammatory reaction that is to be expected during a vaccine response. In conclusion, the systemic no adverse effect level (NOAEL) was determined to be the highest dose tested, 200 μg of ACT-1239.

## 4. Discussion

Malaria remains a leading cause of morbidity and mortality in much of the developing world, accounting for several hundred thousand deaths each year, especially among children [1]. *Plasmodium falciparum*, the predominant species of the pathogen causing malaria in Africa and the leading cause of human infection and death, has a complex life cycle within the mammalian host that presents several opportunities for prophylactic intervention, including the pre-erythrocytic stage that is initiated by the injection of sporozoites by the *Plasmodium*-infected mosquito as it takes a blood meal. Pre-erythrocytic vaccines aim to prevent the development of the blood-stage parasites responsible for clinical disease by targeting both the extracellular sporozoite and the intracellular hepatic stages, both of which express circumsporozoite (CS) protein, which contains several epitopes (T1, B, and T*) that show promise as vaccine components [26,27,28]. Several pre-erythrocytic vaccine candidates are currently in various stages of preclinical and clinical development, including whole attenuated sporozoites [29,30,31] and subunit-based virus-like particles (VLP) [9,11,32,33,34,35]. The most advanced malaria subunit vaccine is RTS,S or Mosquirix™, which presents CS epitopes on hepatitis B surface antigen (HBsAg) virus-like particles and has been approved for use by European regulators [36,37]. RTS,S administered in a potent adjuvant formulation reduced the risk of clinical disease in infants aged 5–17 months by 56% and in neonates aged 6–12 weeks by 31% [8,9]. These studies demonstrate the feasibility of CS subunit vaccines in eliciting protection in the target population, infants, and children <5 years of age, who suffer the majority of deaths due to *P. falciparum* infection. However, RTS,S-induced immune responses were suboptimal as sterile immunity was not obtained and recent follow-up studies indicate that protection wanes rapidly over time in areas of high transmission [10,11,32,38]. R21/Matrix-M, a second CS VLP based on HBsAg, has been shown to be more potent and perhaps more efficacious than RTS,S [34,35] and has recently been endorsed by the World Health Organization. Although whole sporozoite vaccines elicit sterile immunity, this approach faces several logistical hurdles, including the use of human blood and serum, the challenge of growing sporozoites in vitro (although progress has been made in this regard [39]), limited capacity for scale-up, cold chain storage requirements, and the failure to elicit protection when delivered by intradermal, subcutaneous, or intramuscular injection [5,6,7]. Thus, there remains a need to develop novel delivery platforms for pre-erythrocytic malaria vaccines.

We previously reported the application of layer-by-layer fabrication of synthetic microparticles (LbL-MP) to generate a novel pre-erythrocytic malaria vaccine presenting subunit epitopes of the *Plasmodium* CS protein [14]. The initial studies were limited to examination of immunogenicity and efficacy in a mouse challenge model. These efforts have been extended to include additional preclinical species (NHPs and rabbits), modifications to the LbL-MP design, including incorporation of an innate immune agonist, and a demonstration of safety and immunogenicity in a GLP-compliant rabbit toxicology study in preparation for clinical evaluation. The first immune effector mechanism identified in sporozoite-immunized hosts was sporozoite-neutralizing antibodies that target the NANP tetramer repeats in the CS central region [40,41,42,43], immobilizing sporozoites in the skin at the site of the mosquito bite and delaying their egress into the circulation [44], thus providing a longer window for an antibody-mediated attack [45,46,47] and blocking invasion of the host hepatocytes [21,48,49,50]. Induction of high levels of sporozoite-neutralizing antibodies is T-cell-dependent, requiring CD4+ T-cell help for B-cell differentiation and for the development of CD8+ memory T-cells [15,16]. T-cell cytokines such as IFNγ can also directly mediate protective immunity by inhibiting the development of intracellular exoerythrocytic forms (EEF) [17,18]. Analysis of PBMC and T-cell clones from volunteers immunized with irradiated sporozoites revealed two important Class II-restricted epitopes. T1, located in the *P. falciparum* CS minor repeat region comprising alternating NANPNVDP repeats [51], provides an attractive vaccine component as it is conserved in all *P. falciparum* strains. A second epitope, T* (EYLNKIQNSLSTEWSPCSVTSGNG), is located in the C-terminus of *P. falciparum* CS and overlaps the highly conserved region RII-plus that functions in the sporozoite/host cell interaction in the liver [52,53]. T* is a universal T-cell epitope that binds to multiple DR and DQ molecules and was immunogenic in multiple inbred strains of mice [54]. Thus, vaccine candidates were designed to include the antibody (NANP repeats) and T-cell (T1 and T*) epitopes in the designed peptide on the outermost layer of the microparticle.

Building off the prototype candidates previously reported [14], we designed four modified constructs that differed in crosslinking of the homopolymer base layers in the LbL film and the inclusion of Pam3Cys, a TLR2 agonist that serves to engage the innate immune system [55]. Immunogenicity analyses revealed that either single modification improved the potency of the vaccine candidate as evidenced by higher and more persistent parasite-neutralizing antibody responses (Figure 1, Figure 2 and Figure 3), while the double-modified candidate also steered the T-cell response toward a Th1 bias (Figure 1), similar to results we obtained in a respiratory syncytial virus model [56]. The favorable antibody and T-cell responses were further explored in macaques since this species more closely models the human immune response and has been used extensively to study vaccine candidates for malaria [23,57] and other infectious diseases [58,59]. These results confirmed that the base layer-crosslinked, Pam3Cys-modified candidate yielded parasite-neutralizing antibody responses that were protective against the *Plasmodium* challenge following passive administration to naïve mice (Figure 4), and T-cell responses that favor IFNγ over IL-5 (Figure 5). Although the NHP study was performed only once due to logistical and cost limitations, the results clearly demonstrate induction of superior parasite-neutralizing antibody responses in the group immunized with ACT-1201 (bXL, Pam3Cys) compared to ACT-1200 (bXL), thus justifying the selection of the bXL Pam3Cys architecture for final candidate optimization and selection (Figure 4C).

Based on the results of the mouse and macaque studies, the Pam3Cys-modified design was chosen for the development of a pre-erythrocytic malaria vaccine candidate. The final structural point to be examined focused on the single unpaired cysteine residue in the T* epitope, which is subject to unintended and uncontrolled inter-chain disulfide bond formation during peptide synthesis and purification. Two conservative substitutions, C → A and C → S, were tested in parallel candidates, each with and without base layer crosslinking. While both substitutions elicited equivalent antibody responses and protection from the challenge in mice, the C → A candidates failed to prime for T-cell responses detectable in ELISPOT regardless of the crosslinking status (Figure 6). Although the dominant mechanism of protection for pre-erythrocytic vaccines is antibody-mediated neutralization of the sporozoite before it reaches the liver [21,48,49,50], and our own results demonstrate that both candidates provided equal protection from liver-stage infection in mice, T-cells do play an important role in supporting the antibody response and directly killing infected hepatocytes preventing release of blood-stage merozoites [60,61,62], demonstrating that the C → S substitution is favorable. In a final mouse study, the C → S candidates were compared for induction of persistent immunity and efficacy. These results clearly demonstrated that the base layer-crosslinked candidate elicited longer-lasting protection against the challenge, although the efficacy of either candidate was restored by a boost immediately prior to the challenge (Figure 7). Based on the cumulative results of the mouse and NHP studies, ACT-1239 (bXL, Pam3Cys, C → S) was selected for further study and development of a novel vaccine candidate. In a GLP-compliant rabbit toxicology study, four doses of 200 µg ACT-1239 proved to be safe, well-tolerated, and immunogenic, yielding antigen-specific antibody responses (Figure 8) that neutralized the parasite in an in vitro hepatocyte invasion assay (Figure 9), thus identifying the NOAEL and establishing dose and regimen parameters for clinical evaluation of this novel LbL-MP pre-erythrocytic vaccine candidate.

The vital role CS plays in eliciting protective immunity as a stand-alone vaccine [63], in combination with additional sporozoite antigens [64], or in a heterologous prime-and-trap vaccination regimen with whole-organism sporozoites [57,65,66], validates the CS protein as a viable target antigen for malaria vaccine development. However, these studies also demonstrate the need for efficacious subunit vaccines that overcome the problems of scale-up, adjuvant formulation, and cellular immunity experienced in previous nonclinical and clinical studies of CS-based vaccines. The LbL-MP platform addresses each of these challenges: the candidate is manufactured by a synthetic process that is amenable to scale-up; it incorporates a TLR-2 agonist into the particle, thus improving potency by direct engagement of the host’s innate immune system without a complex adjuvant formulation; it elicits long-lasting sporozoite-neutralizing antibodies and Th1-biased T-cell responses; and it provides protection from challenges with live parasites. The current preclinical immunogenicity, efficacy, and safety data validate the selection of ACT-1239 for clinical evaluation as a pre-erythrocytic malaria vaccine candidate.

## 5. Conclusions

The CS protein of *P. falciparum* has been validated as a pre-erythrocytic malaria vaccine target antigen due to its important role in the parasite infectious cycle and the ability of CS-specific antibody responses to protect the host from infection. Subunit epitopes (antibody target B and T-cell targets T1 and T*) of CS were engineered into synthetic microparticle vaccines made by layer-by-layer fabrication and tested for induction of protective antibody responses in mice and NHPs. While the initial results were positive, several modifications to the vaccine candidate improved its potency and manufacturability. The inclusion of a TLR2 ligand to engage the host’s innate immune response increased the potency and efficacy of the candidates, as did crosslinking of the particle base layer film prior to the addition of the target antigen. Finally, the replacement of an unmatched cysteine with a serine residue simplified the manufacturing process while retaining potency in mice and NHPs. The base layer-crosslinked, Pam3Cys-modified, serine-substituted vaccine candidate was safe and immunogenic in a GLP-compliant toxicology study, thus demonstrating its suitability for a clinical evaluation of its safety and potency.

## Figures and Tables

**Figure 1 vaccines-11-01789-f001:**
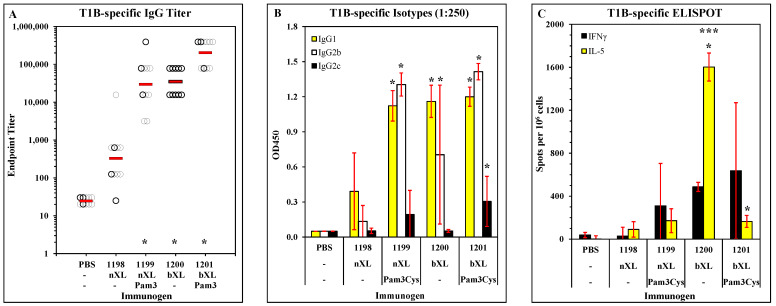
The immune response elicited by immunization with T1BT* microparticles. Mice were immunized with the indicated treatments on days 0, 21, and 42. (**A**) Sera collected on day 49 were assayed in ELISA against T1B peptide. Results show endpoint titers of individual sera (open circles) and the group geometric mean titers (red bars). * *p* < 0.05 compared to the PBS group. (**B**) Sera were diluted 1:250 and assayed against T1B peptide and then probed with isotype-specific detection antibodies. Results show mean ± SD of 10 mice per group. * *p* < 0.05 compared to the same isotype in the PBS group. (**C**) Spleen cells were harvested from 3 mice per group on day 49, lymphocytes were restimulated with T1B peptide in IFNγ and IL-5 ELISPOT plates, and the number of spot-forming cells was counted in an AID ViruSpot Reader. Results show mean ± SD of 3 mice per group. Groups were compared by Student’s *t*-test. * *p* < 0.05 compared to the PBS group. *** *p* < 0.05 compared to 1201 group.

**Figure 2 vaccines-11-01789-f002:**
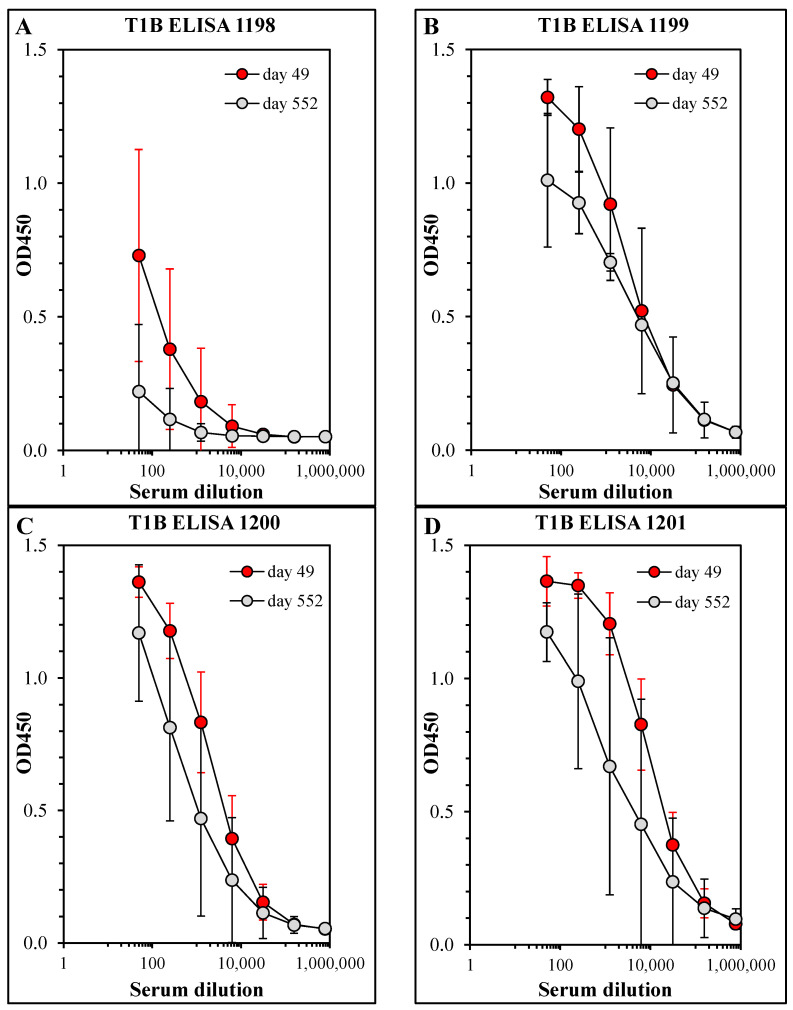
Persistence of the mouse antibody response elicited by immunization with the LbL-MP vaccine. Mice from Figure 1 were maintained without further manipulation until day 552, when they were bled by a retroorbital puncture. Sera collected on day 49 (red symbols) and day 552 (gray symbols) were assayed in ELISA against T1B peptide. Results show the mean ± SD of 10 mice per group (closed symbols) or 8 mice per group (open symbols). (**A**) ACT-1198 mice. (**B**) ACT-1199 mice. (**C**) ACT-1200 mice. (**D**) ACT-1201 mice.

**Figure 3 vaccines-11-01789-f003:**
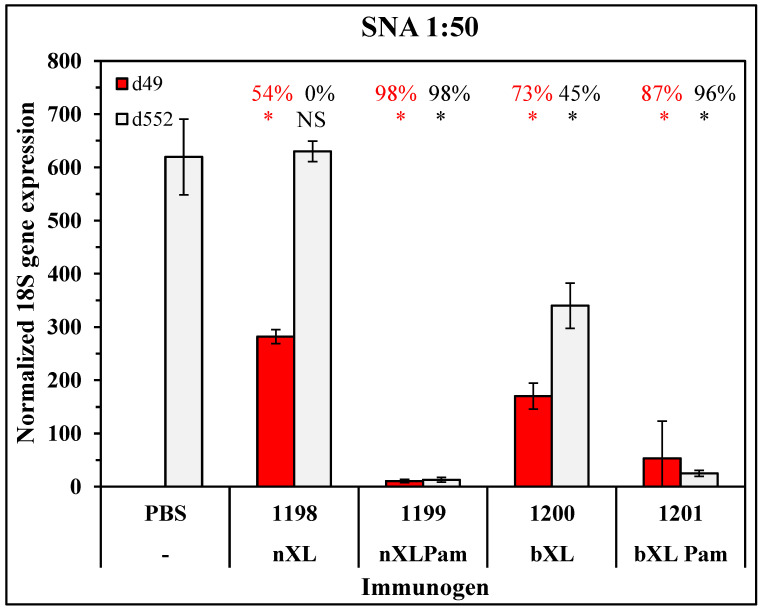
Sporozoite-neutralizing antibody activity. Sera from Figure 2 were pooled by group, diluted 1:50, and pre-incubated with Pf sporozoites prior to the addition to HepG2 cells. After 3 days of culture, RNA was harvested and Pf 18S rRNA was quantitated by qPCR and normalized against human β-actin expression. Results show 18S rRNA copy number (normalized against actin) for pooled mouse sera (mean ± SD of duplicate wells per sample). Insets show a % reduction in 18S signal compared to the PBS mock-immunized serum values; * *p* < 0.05 compared to the PBS group (NS = not significant).

**Figure 4 vaccines-11-01789-f004:**
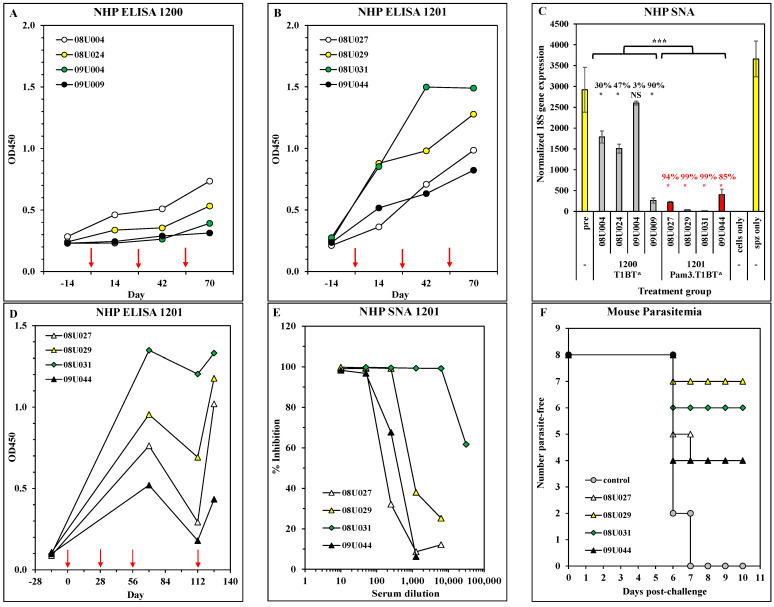
Immunogenicity and efficacy in NHPs. Macaques were pre-bled on day −14, immunized i.d. with 50 µg of ACT-1200 or ACT-1201 on days 0, 28, and 56 (as indicated by ↓), and bled on days 14, 42, and 70 (14 days following each immunization). (**A**,**B**) Sera from each NHP were tested in the T1BT* ELISA. Results show OD450 at a serum dilution of 1:125 for each individual monkey, with monkey identification numbers shown in the legend. (**C**) Day 70 sera from each NHP were tested in the sporozoite neutralization assay. Results show 18S rRNA copy number (normalized against actin) for individual NHP sera assayed in duplicate wells (mean ± SD). pre = mean ± SD of 8 pre-bleed sera. Insets show a % reduction in 18S signal compared to the pre-bleed serum values; * *p* < 0.05 (NS = not significant) for individual sera compared to the pre-bleed serum values; *** *p* < 0.05 between the two groups of animals. (**D**) NHPs were bled again on day 112 (8 weeks post-third immunization), received a fourth immunization on the same day 112 (↓), and were bled a final time on day 126 (2 weeks after the fourth immunization). Sera collected on the indicated days were tested in T1BT* ELISA. Results show OD450 at a serum dilution of 1:125 for each individual monkey, with monkey identification numbers shown in the legend. (**E**) Individual sera collected on day 126 were tested in the sporozoite neutralization assay. Results show titration curves of sera from 4 individual macaques. (**F**) Ig was purified from each individual macaque serum (day 126) and administered to 8 naïve mice per donor macaque prior to the challenge with live Tg-*Pb*/*Pf*CSP parasite, and parasitemia was monitored post-challenge. Results show the number of mice per group that remained parasite-free as determined by microscopic examination of Geimsa-stained blood smears. No further changes were detected up to day 15.

**Figure 5 vaccines-11-01789-f005:**
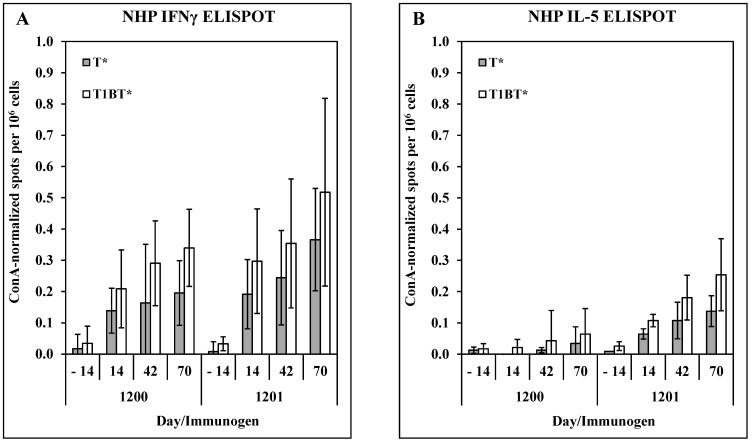
NHP T-cell responses. PBMC collected on the indicated days were tested against T* or T1BT* peptide in IFNγ (**A**) or IL-5 (**B**) ELISPOT assays. Results were normalized to the maximal responses obtained with positive control ConA and show mean ± SD of 4 monkeys per group.

**Figure 6 vaccines-11-01789-f006:**
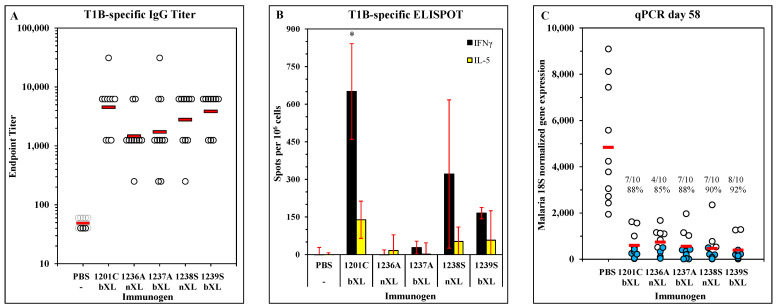
Immunogenicity and efficacy of T1BT* LbL-MP with substitutions for the unpaired cysteine. Mice were immunized on days 0, 21, and 42 with the indicated candidates, including ACT-1201 as the positive control. (**A**) Sera were harvested on day 59 and tested in ELISA against T1B peptide. Results show endpoint titers of individual sera (open circles) and the group geometric mean titers (red bars). All treatment groups were statistically different (*p* < 0.05) from the PBS group but not from one another. (**B**) Spleen cells were harvested on day 59 and restimulated with T1B peptide in IFNγ and IL-5 ELISPOT plates. The data depict the mean ± SD of 3 mice per group. * *p* < 0.05 compared to the PBS group. (**C**) Mice were challenged with PfPb on day 63 and sacrificed 40 h later. The parasite burden in the livers was measured by qPCR. Results show the 18S rRNA copy number for individual mice (circles) and group means (red bars). Insets show the # of mice protected (>90% reduction in 18S rRNA expression compared to PBS mean, indicated by blue-filled circles) and group mean reduction in copy number. All treatment groups were statistically different (*p* < 0.05) from the PBS group but not from one another.

**Figure 7 vaccines-11-01789-f007:**
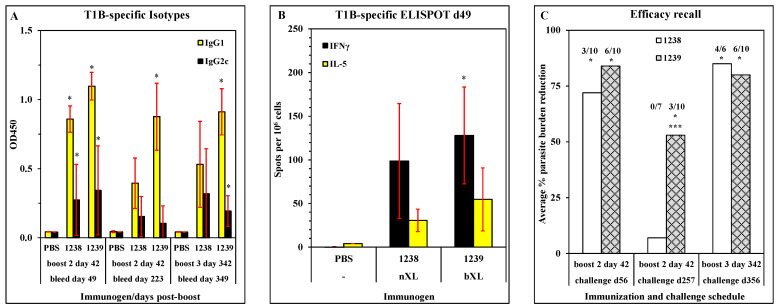
Persistence of antibody responses and efficacy elicited by ACT-1238 and ACT-1239. C57BL/6J mice were immunized on days 0, 21, 42, and 342 as detailed in Table 2. (**A**) Sera were collected on days 49 (7 days post-boost 2), 223 (181 days post-boost 2), and 349 (7 days post-boost 3) and tested in isotype-specific ELISA against T1B peptide. Results show the mean ± SD of 40 mice per group. * *p* < 0.05 compared to the PBS group on the same day, same isotype. (**B**) Spleen cells were harvested on day 49 (7 days post-boost 2) and restimulated with T1B peptide in IFNγ and IL-5 ELISPOT plates. Results show the mean ± SD of 3 mice per group. * *p* < 0.05 compared to the PBS group. (**C**) Mice were challenged with PfPb on days 56, 257, or 356 and sacrificed 40 h later. The parasite burden in the livers was measured by qPCR. Results show a group percent reduction in parasite burden compared to naïve challenged mice at each time point. Insets show the number of individual mice protected (≥90% reduction in parasite burden) over the number of mice challenged in each group. * *p* < 0.05 compared to naïve challenged mice at each time point. *** *p* < 0.05 compared to the 1238 group at the same time point.

**Figure 8 vaccines-11-01789-f008:**
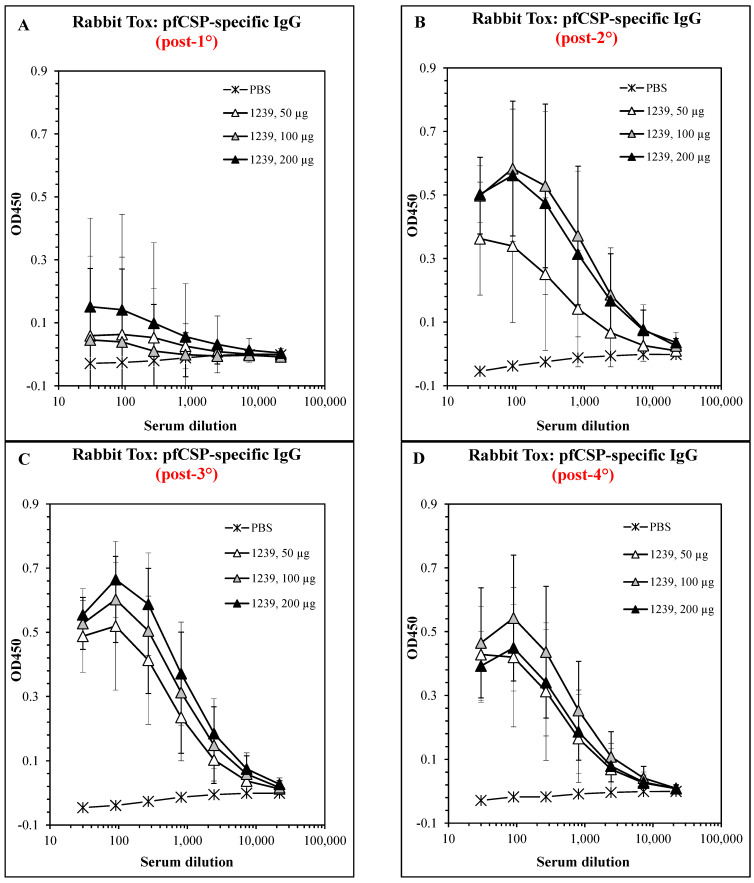
Antibody responses of rabbits immunized with ACT-1239. Rabbits were immunized on days 0, 28, 56, and 84 and bled 7 days after each immunization. Sera were tested in ELISA on PfCS-coated plates. Values obtained with matched pre-immune sera were subtracted to yield peptide-specific responses. (**A**) Day 7 sera. (**B**) Day 35 sera. (**C**) Day 63 sera. (**D**) Day 91 sera. Results show mean ± SD of 12 serum samples per group.

**Figure 9 vaccines-11-01789-f009:**
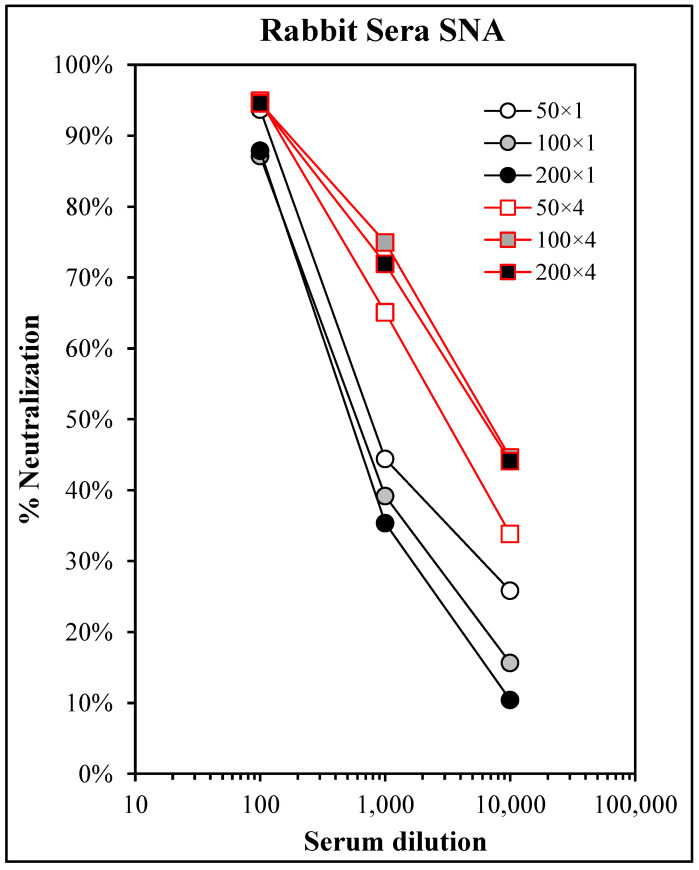
The potency of sporozoite-neutralizing antibodies in rabbits immunized with ACT-1239. Sera were pooled from rabbits immunized with 50, 100, or 200 µg doses of ACT-1239-14-TOX once (×1) or four times (×4), serially diluted and tested in the SNA as described. Results show a % reduction in Pf 18S rRNA signal compared to control wells with pre-immune sera.

**Table 1 vaccines-11-01789-t001:** LbL-MP constructs. bXL = PLL:PGA base layers were chemically crosslinked prior to adding DP to the microparticle vaccine. nXL = base layers were not crosslinked. Pam3Cys = TLR2 ligand covalently linked to DP prior to the addition to the microparticle vaccine. (A) and (S) = unpaired cysteine residue in T* was substituted with alanine or serine, respectively, to prevent inter-chain disulfide bond formation in the DP.

Construct	Crosslinking	DP Shorthand
ACT-1198	nXL	T1BT*
ACT-1199	nXL	Pam3Cys.T1BT*
ACT-1200	bXL	T1BT*
ACT-1201	bXL	Pam3Cys.T1BT*
ACT-1236	nXL	Pam3Cys.T1BT*(A)
ACT-1237	bXL	Pam3Cys.T1BT*(A)
ACT-1238	nXL	Pam3Cys.T1BT*(S)
ACT-1239	bXL	Pam3Cys.T1BT*(S)

**Table 2 vaccines-11-01789-t002:** Timeline and study endpoints for mouse immune persistence study. ELISPOT: mice were sacrificed, and spleen cells were analyzed in IFNγ and IL-5 ELISPOT plates. ELISA: mice were bled by retro-orbital puncture, and sera were analyzed by ELISA against T1B peptide. PfPb: mice were challenged by exposure to bites of PfPb-infected mosquitoes. Liver: mice were sacrificed, and the parasite burden in the liver was measured by qPCR; 4°: mice were given a fourth immunization identical to the first three. Results are shown in Figure 7.

	Day of Study												
*n*	0	21	42	49	56	58	223	257	259	342	349	356	358
3	1°	2°	3°	ELISPOT	-	-	-	-	-	-	-	-	-
10	1°	2°	3°	ELISA	PfPb	liver	-	-	-	-	-	-	-
10	1°	2°	3°	ELISA			ELISA	PfPb	liver	-	-	-	-
10	1°	2°	3°	ELISA						4°	ELISA	PfPb	liver

## Data Availability

All data are reported in the current article.

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
