# Peer review of "Immunogenicity, Efficacy, and Safety of a Novel Synthetic Microparticle Pre-Erythrocytic Malaria Vaccine in Multiple Host Species"

_vaccines, 2023, doi:10.3390/vaccines11121789_

Round 1
Reviewer 1 Report
Comments and Suggestions for Authors
Authors prepared a novel synthetic microparticle pre-erythrocytic malaria vaccine by The LbL-by-TFF method and detected its immunogenicity and efficacy, and safety of in different animals. The results showed higher and persistent antibody titers of the Pam3Cys-modified constructs in mice and in vitro when compared to unmodified prototype one, with higher safety. It is very interesting and important work. Some questions need to resolve before publishing:
Line 148-158 may be combined into Line 159-168, and please give the details of ELISA or SNA, not described below.
Line 241-246 may be removed to Introduction.
In 3.2 Persistence of antibody responses, why do you detect antibody only on day 49 and day 552? What about the titer of antibody at other time? Please explain it or give some references if it is possible.
Please provide the conclusions section.
Other minors:
References 36, 40, 57, etc.
Reviewer 2 Report
Comments and Suggestions for Authors
The authors are reporting eight different types of microparticle (MP)-based vaccine formulations as vaccine candidates against the per-erythrocyte parasite form of the malaria pathogen Plasmodium falciparum. These vaccine candidates are intended to block the invasion of liver cells (hepatocytes) by the sporozoite form of the pathogen, that is released into the bloodstream when an infected female Anopheles mosquito feasts on a human host. The work presented is very impressive and extensive. The microparticles are fabricated on a calcium carbonate core with a layer-by-layer deposition approach of seven negatively or positively charge polypeptide layers, i.e. poly-L-glutamic acid (PGA), poly-L-lysine (PLL), after the antigen, the “designed peptide” (DP) was coated onto the multilayered MP. Before DP coating, the layers were either crosslinked with EDC or not crosslinked, and different versions of the DP were deposited. DP contained a B and T cell epitope sequence of the P. falciparum circumsporozoite protein, CS. The latter was provided either alone or as a conjugate with the TLR2 ligand Pam3Cys, and with or without the exchange of the only cysteine in the sequence to alanine or serine to prevent disulfide bond formation. Vaccine candidate formulations were tested in three different mamalian species, mice, NHP, and rabbits, the latter supposedly for toxicology studies. Mice were immunized via the foot pad and analyzed for IgG production, IgG isotype generation, as well as in IL-5 and IFN-gamma ELISPOTS. Furthermore, the authors show that all but one non-crosslinked MP vaccine formulation generated long-term antibody responses > 552 days. The mouse antisera from the vaccine formulations were also tested in sporozoite neutralization activity (SNA) with cultured human HepG2 hepatoma cells (hepatocytes). Next, malaria naïve female rhesus macaques (NHP model) were used to test the two best performing vaccine candidate MP formulations with crosslinked base layers, without (ACT-1200) and with Pam3Cys conjugation (ACT-1201), by measuring vaccine induced ELISA antibody titers and SNA. Furthermore, antibodies from vaccinated NHPs were tested in a passive vaccination mouse model with CS expressing transgenic mouse-specific P. berghei malaria parasites. Finally, a rabbit model was use for toxicology studies of the vaccine candidates, and efficacy recall assays were conducted in mice.
Overall, the study presented is impressive and rich in supporting data, a few points of critique should be considered for improvement: It would be useful if the authors could better define what the difference between the DP, T1BT* and T* peptide or T* epitope is. This could easily be done by clarifying or annotating the DP sequence (Line 125). Currently this part is very confusing.
Furthermore, it is odd that none of the SNA assays in mice or NHPs shows any error bars or standard deviations. Were technical replicates or repeats of the experiments not performed? Likewise, it is difficult to understand why the 1:10 dilutions of the antisera in Figure 3 B and D at day 552 produce considerably less of a neutralizing effect than the 1:50 dilutions. Could this be due to large errors or variations in the measurements?
For the rabbit safety toxicology study only immune responses were reported, but no actual toxicology data were provided. One would expect at least liver and kidney function panels. There is no discussion the tolerance to and fate of the CaCO3 crosslinked poly-L-Glu, poly-L-Lys, MPs. The reader wonders what pharmacological effects and MP degradation pathways to expect from the vaccine constructs.
While the role of T cells is discussed as having an enhancing effect on antibody production, it is unclear what the relevance of CD8+ T cells (Line 83) should be. Would they recognize and kill infected hepatocytes?
Other major points are:
Figures of sporozoite neutralization assays (SNA): Please present the results of the controls: HepG2 cells + Spz and HepG2 cells in all Figures on SNA.
Lines 345-348: The authors performed T1B ELISA assays to analyze sera from non-human primates before and after vaccination with ACT-1200 or ACT-1201, obtaining the data presented in Figure 4 A, B. The data belonging to group ACT-1200 showed that NHP #09U009 has almost the lowest antibody response. However, the statement on lines 345-348 says that the results obtained in the sporozoite neutralization assay (SNA) have the same trend as the ELISA results, where the antiserum of NHP #09U009 immunized with ACT-1200 has a high Pf neutralizing activity. Therefore, ELISA and SNA assays in non-human primates immunized with 1200 do not show the same general trend as the authors claim.
Figure 5 A, B: The non-human primates were immunized on days 0, 28 and 56 and bled 14 days following each immunization to isolate sera and PBMC to evaluate de immunogenicity. However, the authors present the results of IFN-γ and IL-5 ELISPOT measured on days -14, 14, 42 and 56 instead of days -14, 14, 42 and 70 in Figure 5.
Line 358: Please mention why the IFN-γ and IL-5 responses of the fourth immunization to the ACT-1201 group were not evaluated.
Figure 7 and lines 488-490: Why did the authors not include ACT-1201 as a control for the further evaluation of the vaccine candidates 1238 and 1239? In the previous experiments, seems that 1201C had a superior antibody and T cell responses compared with ACT-1239, so it would be interesting to compare with the candidate without the Cysteine mutation to finally make a more accurate conclusion about the superior immunogenicity of ACT-1239 and select it as the final candidate eliminating the manufacturing risk by the substitution of the unpaired Cysteine.
According to the results presented, the candidate vaccine ACT-1239 does not have 100% efficacy, which means that some sporozoites will complete the infection of the hepatocytes and initiate a blood-stage infection, continuing the life cycle of the parasite. Please consider discussing the importance of multi-epitope and multi-stage malaria vaccines.
The study did not include controls using LbL-MP linked with and without Pam3Cys to determine if the antibody and T-cell responses were specific or not to the designed circumsporozoite peptide.
Minor points:
Please check Lines 92-93. The sentence is confusing and states exactly the opposite of what was done. It should be substitution of the unpaired Cys with Ser, not the other way around.
Line 99: Non-human primates are mentioned for the first time on this line and the abbreviation NHP is mentioned on line 335. Please mention the abbreviation where it first appears in the text.
Line 217: Remove the underline from the word “from”.
Line 351: Passive immunization of mice (PIM) was already abbreviated in line 228.
Line 434: Please clarify in the legend that in Figure 6A “1201C” refers to the control group.
Line 463: Verify if the correct day on which efficacy was monitored was 212 or 257, and correct Table 2 and Figure 7 accordingly.
Line 553: Please mention the abbreviation circumsporozoite (CS) where it first appears in the text (line 61).
Reviewer 3 Report
Comments and Suggestions for Authors
The study (Vaccines-2705946) developed a synthetic microparticle vaccine for malaria using layer-by-layer fabrication and tested its immunogenicity, efficacy, and safety in mice and non-human primates. The topic is interesting but the authors need to improve the quality before publishing as follows:
1.The abstract is not well written, it is difficult to understand objectives, which experiments did the authors perform.
2.The authors need to detail where materials and animals were purchased and processed before conducting experiments.
3.How do the authors define parasite free? Please detail the method of parasite counting.
4.Most of the experiments are in triplicate, did the authors repeat in another independent experiment?
5.Data analysis: please fully describe the stats analysis, why a paired t-test was used? Data presented in figures are not normally distributed, a non-parametric test is more appropriate.
Did the author compare multiple groups?
6.Figures: the authors need to describe in more detail, so readers can understand the figures without reading the manuscript. You also need to indicate what * is presented in the figure and which groups do you compare?
7.Fig. 4. What do circles mean?
8.IRB ID number should be provided
9.The paper does not provide information about the specific limitations or challenges encountered during the development and evaluation of the synthetic microparticle vaccine.
10.A conclusion section is missing
Round 2
Reviewer 3 Report
Comments and Suggestions for Authors
The authors address many of the comments raised, but they do not fully address some of them as follows:
3. Parasite Detection Method "parasite-free"
The author mentions that the number of mice per group that remained parasite-free was determined by microscopic examination of Giemsa-stained blood smears. However, the statement is not clear about the details of the examination. It is unclear whether the examination was performed on a thick film or a thin film of Giemsa-stained blood smears. Moreover, the statement does not specify the number of fields examined to determine the parasite-free status of the mice.
A thick film of Giemsa-stained blood smear is preferred over a thin film for detecting low levels of parasitemia. The number of fields examined should be sufficient to detect any parasites present in the sample. Which could be an objective measurement when the researchers were not blinded. These details are important to ensure the accuracy and reliability of the results.
4. Replication of Results
The authors need to discuss why they did not repeat the animal experiment as a limitation in the discussion section. If other researchers cannot duplicate their results, there is a reason for that.
